# Majorana edge states in Kitaev chains of the BDI symmetry class

Anton Bespalov

Institute for Physics of Microstructures, Russian Academy of Sciences,
603950 Nizhny Novgorod, GSP-105, Russia
National Research Lobachevsky State University of Nizhny Novgorod,
603950 Nizhny Novgorod, Russia

## Abstract

Majorana edge states in Kitaev chains possessing an effective time reversal symmetry with one fermionic site per unit cell are studied. For a semi-infinite chain the equations for the wave functions of Majorana zero modes can be reduced to a single Wiener-Hopf equation, which has an exact analytical solution. We use this solution to determine the asymptotic behaviors of the Majorana wave functions at large distances from the edge of the chain for several infinite-range models described in the literature with focus on a model with slow power-law falloff of pairing and hopping amplitudes. For these models we also determine the asymptotic behavior of the energy of the fermionic state composed of two Majorana modes in the limit of long (finite) Kitaev chains.

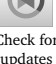
# 1 Introduction

Topological superconductors offer a promising platform for the implementation of a topological quantum computer [1,2]. The simplest generic model of a topological superconductor has been proposed by Alexei Kitaev [3]. He considered a chain of spinless fermions with $p$-wave superconducting pairing – this model became famous as the Kitaev chain. Such chain with one fermionic site per unit cell is described by the Hamiltonian

$$\hat{H} = \frac{1}{2} \sum_{l,n} \left[ 2t_{l-n} \hat{a}_l^\dagger \hat{a}_n + \Delta_{l-n} \hat{a}_l^\dagger \hat{a}_n^\dagger + \Delta_{l-n}^* \hat{a}_n \hat{a}_l \right]. \tag{1}$$

Here, the indices $n$ and $l$ number the fermionic sites, $t_l$ are hopping amplitudes, $\Delta_l$ are pairing amplitudes, the star $*$ denotes complex conjugation, $\hat{a}_l^\dagger$ and $\hat{a}_l$ are fermionic creation and annihilation operators, respectively: $\{\hat{a}_l, \hat{a}_n^\dagger\} = \delta_{ln}$, $\{\hat{a}_l, \hat{a}_n\} = 0$. The paring and hopping amplitudes satisfy the relations $t_{-l} = t_l^*$, $\Delta_{-l} = -\Delta_l$. Under certain conditions, a semi-infinite Kitaev chain ($l = 0, 1, 2...$) hosts a so-called Majorana zero mode (MZM) – a special kind of edge state with energy exactly equal to zero. In this case, the system is said to be a 1D topological superconductor. In a finite chain, MZMs appear at both ends of the chain, and they combine into an ordinary Bogoliubov quasiparticle possessing a near-zero energy $E_0$. MZMs are exemplary non-Abelian anyons, which are the key component of a topological quantum computer.

Despite the apparent artificiality of Kitaev's model, it had a huge impact on condensed matter physics and materials science. Kitaev's work has stimulated the experimental search of MZMs in quantum wires with induced effective $p$-wave pairing [4], and has inspired a vast amount of theoretical research. Most proposals for implementations of 1D topological superconductors are rather described by a continuous version of the Hamiltonian (1), however, some system based on atomic chains placed on superconductors have effective discrete Hamiltonians that are identical to (1) [5–7]. In such systems the spatial structure (probability density) of MZMs can be probed using scanning tunneling microscopy [8]. The Hamiltonian (1) and its 2D analog are also relevant for the description of cold atoms in optical lattices [9–12].

To determine the quasiparticle spectrum of the Kitaev chain, one should apply a Bogoliubov transformation that diagonalizes the Hamiltonian (1) [9]. This results in the following Bogoliubov-de Gennes (BdG) equations for quasiparticle wave functions $(u_l, v_l)^T$:

$$\sum_n \begin{pmatrix} t_{l-n} & \Delta_{l-n} \\ \Delta_{n-l}^* & -t_{n-l} \end{pmatrix} \begin{pmatrix} u_n \\ v_n \end{pmatrix} = E \begin{pmatrix} u_l \\ v_l \end{pmatrix}, \tag{2}$$

where $E$ is the quasiparticle energy. To obtain the structure of an isolated MZM, we have to put $E = 0$ and consider a semi-infinite chain: $l, n = 0..\infty$. In the case of short-range hopping and pairing – $t_l = \Delta_l = 0$ for $|l| > 1$ – the MZM has been analyzed in detail by Kitaev [3], who found that its wave function falls off exponentially fast with distance from the chain edge. This result is valid for any model with finite-range couplings: one can prove this be seeking the solution of Eq. (2) in the form of a sum of exponentially decaying terms: $u_l, v_l \propto e^{-\lambda l}$ (possibly, with polynomial factors). This standard method for solving systems of recurrence relations is sometimes referred to as the scattering approach in physical literature [13]. Equation (2) within finite-range models also can be solved using the transfer-matrix approach [14, 15]. The situation changes drastically when infinite-range hopping or pairing is present. In particular, it has been demonstrated that in the case of a power-law falloff of hopping and pairing amplitudes – $t_l \propto |l|^{-\alpha}$, $\Delta_l \propto |l|^{-\beta}$, $\alpha, \beta \geq 1$ – the MZM wave function also falls off in a power-law manner with increasing $l$ [5, 16–20].

Formally, Eq. (2) on a semi-infinite chain constitutes a vector Wiener-Hopf problem [21, 22]. To date, a general solution method for this problem is unknown even for two-component

vectors. In practice, the wave functions of MZMs in chains with infinite-range couplings are determined numerically by solving Eq. (2) on a finite chain (unless a solution for a semi-infinite system can be guessed, like in Ref. [16]). However, it turns out that for a wide class of systems an exact analytical solution of our Wiener-Hopf problem can be obtained.

First, it was noticed by Bespalov [7] that in a special case for $E = 0$ the BdG equations decouple, and two scalar Wiener-Hopf problems arise, which can be solved analytically using Gakhov's approach [23]. Recently, it has been realized that the same trick is applicable when the coefficients $t_l$ and $\Delta_l$ are real [20]. It is said then that the system possesses an effective time-reversal symmetry and belongs to the Altland-Zirnbauer class BDI [24–26]. If we put $E = 0$ in Eq. (2), we may find that the equations for $u_l + v_l$ and $u_l - v_l$ decouple, so that the BdG equations can be solved exactly. Using the Wiener-Hopf factorization, the authors of Ref. [20] performed an exhaustive study of MZM wave functions and of the energies of Majorana edge modes in long chains for power-law models with exponents $\alpha > 1$ and $\beta > 1$. Apparently, their approach based on manipulations with absolutely summable Laurent series can not be directly extended to the range of parameters $\alpha \leq 1$ and $\beta \leq 1$. This range is, in fact, of great fundamental interest: there, the half-chain entanglement entropy exhibits an unusual logarithmic scaling with chain length [17, 18, 27, 28].

In the present paper, within the Wiener-Hopf approach, we develop an original technique to calculate the asymptotic behaviors of MZM wave functions in the limit of large $l$ for Kitaev chains belonging to the BDI symmetry class. The technique is based on Gakhov's formulas for Wiener-Hopf factors [Eqs. (13) and (14)]. It allows us to reproduce known results for the power-law model with $\alpha > 1$ and $\beta > 1$, as well as to investigate the parameter range $\alpha < 1$ and $\alpha \leq \beta$. There, we find that the MZM wave functions exhibit a power-law falloff. In particular, $u_l, v_l \propto l^{(\alpha-3)/2}$ for $\alpha < \beta$. Remarkably, the decay rate of the MZM wave function decreases with increasing $\alpha$. We observed this feature using numerical solutions of the BdG equations in long open chains. However, the power-law behavior of the wave functions is not clearly observable in chains containing up to 20000 sites.

A crucial characteristic of a Majorana qubit is the energy $E_0$ of the fermionic edge state composed of two MZMs. In finite Majorana wires (and Kitaev chains) this energy is generally non-zero, and it determines the characteristic time $\hbar/E_0$ during which a phase error occurs in an isolated qubit [3]. A part of the present paper is devoted to the discussion of the asymptotic behavior of the energy $E_0$ in the limit of long chains. We demonstrate how this behavior can be determined using the obtained asymptotics for the MZM wave functions, and we confirm our results for the power-law model with $\alpha < 1$ and $\alpha < \beta$ using numerical calculations.

In addition to the power-law model, we study a model with exponential falloff of hopping amplitudes and nearest-neighbor pairing. There, in a natural way, the MZM wave functions exhibit exponential behavior.

The paper is organized as follows. In Sec. 2, we provide a relatively simple solution for the MZM wave functions in a semi-infinite chain using the Wiener-Hopf approach. In Sec. 3, we consider Kitaev chains with power-law falloff of pairing and hopping amplitudes. Sec. 4 is devoted to a model with exponential falloff of hopping amplitudes. In Sec. 5, the asymptotic behavior of the energy $E_0$ is analyzed in the limit of long (finite) Kitaev chains. In the conclusion, the main results are summarized. The appendices contain some technical details.

## 2 General solution for Majorana edge modes

Here and further we will consider the BdG equations (2) with real coefficients $t_l$ and $\Delta_l$. Let us calculate the wave functions of MZMs in a semi-infinite chain, such that $l, n = 0, 1, 2....$ If we put $E = 0$ and introduce a new set of unknown coefficients $s_l = u_l + v_l$ and $w_l = u_l - v_l$,

we obtain the equations

$$\sum_{n=0}^{\infty}(t_{l-n}+\Delta_{l-n})s_n = 0\,, \tag{3}$$

$$\sum_{n=0}^{\infty}(t_{l-n}-\Delta_{l-n})w_n = 0\,. \tag{4}$$

Now we are dealing with scalar Wiener-Hopf equations. They can be solved using the method found by Gakhov [23], which is outlined below. For a special case, the method was previously described in Ref. [7], and the solution in full generality can be also found in the recent paper [20].

For any square-summable solution of Eq. (3) we define its generating function $s(z)$:

$$s(z) = \sum_{l=0}^{+\infty} s_l z^l\,. \tag{5}$$

Generally, we know only that the series converges for $|z| < 1$. Let us define one more generating function:

$$Q(z) = \sum_{l=-\infty}^{+\infty}(t_l + \Delta_l)z^l\,. \tag{6}$$

For a start, we will assume that $t_l$ and $\Delta_l$ fall off exponentially fast with increasing $l$. Then, $Q(z)$ is regular in a ring $R_0 = \{z | \rho_1 < |z| < \rho_2\}$, where $\rho_1 < 1 < \rho_2$. We may note that the function $Q(z)$ defines the bulk spectrum of an infinite chain. Indeed, if we put in Eq. (2) $u_l, v_l \propto e^{ikl}$, where $k$ is a dimensionless quasi-wavenumber, we find the quasiparticle energies

$$E(k) = \pm\left|Q(e^{ik})\right|\,. \tag{7}$$

For the existence of localized zero-energy modes, the bulk spectrum must be gapped, i.e., $Q(z) \neq 0$ on the unit circle. Then, a ring $R = \{z | r_1 < |z| < r_2\}$ containing the unit circle exists, where both $Q(z)$ and $Q^{-1}(z)$ are regular. For $r_1 < |z| < 1$ the product $Q(z)s(z)$ is regular, and the series in Eqs. (5) and (6) can be multiplied in the standard way:

$$Q(z)s(z) = p(z)\,, \tag{8}$$

where, due to Eq. (3), the series for $p(z)$ contains only negative powers of $z$:

$$p(z) = \sum_{l=-\infty}^{-1} p_l z^l\,. \tag{9}$$

It follows from this that $p(z)$ can be analytically continued to the region $|z| > r_1$. In the ring $R$ we have

$$s(z) = Q^{-1}(z)p(z)\,, \tag{10}$$

so that $s(z)$ in our case is regular for $|z| < r_2$. Since $r_2 > 1$, this means that the coefficients $s_l$ fall off exponentially fast with increasing $l$ – this is the first non-trivial result that we obtain.

Now we will determine all functions $s(z)$ and $p(z)$ that satisfy Eq. (8) in the ring $R$. To solve this problem, we make use of the factorization [23]

$$Q(z) = Q_-(z)z^{\kappa}Q_+(z)\,, \tag{11}$$

where $Q_+(z)$ and $Q_+^{-1}(z)$ are regular for $|z| < r_2$, $Q_-(z)$ and $Q_-^{-1}(z)$ are regular for $|z| > r_1$ (including $z = \infty$), and $\kappa$ is an integer, which is called the Cauchy index (winding number) of $Q(z)$. It is given by

$$\kappa = \frac{1}{2\pi} \oint_{|z|=1} d(\arg Q(z)). \tag{12}$$

Note that $\kappa$ coincides up to sign with the winding number defined in Ref. [14], and thus it is the $\mathbb{Z}$ topological invariant of our system. The functions $Q_+(z)$ and $Q_-(z)$ can be taken in the form

$$Q_+(z) = \exp\left( \frac{1}{2\pi i} \oint_{|t|=1} \frac{\ln(Q(t)t^{-\kappa})}{t-z} dt \right), \tag{13}$$

for $|z| < 1$, and

$$Q_-(z) = \exp\left( -\frac{1}{2\pi i} \oint_{|t|=1} \frac{\ln(Q(t)t^{-\kappa})}{t-z} dt \right), \tag{14}$$

for $|z| > 1$. The choice of the branch of the logarithms does not matter here. In the ring $R$, the functions $Q_+(z)$ and $Q_-(z)$ are uniquely defined up to constant numerical factors.

From Eqs. (8) and (11) we obtain

$$Q_+(z)s(z) = z^{-\kappa}Q_-^{-1}(z)p(z). \tag{15}$$

Two cases are possible. If $\kappa \geq 0$, the Laurent series of the right-hand side of Eq. (15) contains only negative powers of $z$, while the Laurent series of the left-hand side contains only non-negative powers of $z$, which means that both sides must be equal to zero. We find then that $s(z) = 0$, and $s_l = 0$ for all $l$. The situation is different for $\kappa < 0$. We have then

$$Q_+(z)s(z) = P_{|\kappa|}(z) + \tilde{p}(z), \tag{16}$$

where

$$P_{|\kappa|}(z) = \sum_{m=0}^{|\kappa|-1} a_m z^m. \tag{17}$$

$a_m$ are some (arbitrary) constants, and the function $\tilde{p}(z)$ is a function whose Laurent series contains only negative powers of $z$. On the other hand, the Laurent series of $Q_+(z)s(z)$ contains only non-negative powers of $z$, which means that $\tilde{p}(z) = 0$, and

$$s(z) = Q_+^{-1}(z)P_{|\kappa|}(z). \tag{18}$$

The expansion coefficients of $s(z)$ are given by the relation

$$s_l = \frac{1}{2\pi i} \oint_{|z|=1} \frac{s(z)dz}{z^{l+1}} = \frac{1}{2\pi i} \oint_{|z|=1} \frac{P_{|\kappa|}(z)dz}{Q_+(z)z^{l+1}}. \tag{19}$$

We have demonstrated that any square-summable solution of Eq. (3) can be written in the form (19). On the other hand, we can do the calculations in reverse order to prove that any sequence $s_l$ given by Eq. (19) satisfies Eq. (3). Thus, Eq. (19) provides a general square-summable solution to Eq. (3). There are overall $|\kappa|$ linearly independent solutions for negative $\kappa$.

Equation (4) can be solved in a similar way. The difference is that we arrive at a relation similar to Eq. (8) with the function $Q(z^{-1})$ in the place of $Q(z)$. The Cauchy index of $Q(z^{-1})$ equals $-\kappa$. Hence, for $\kappa > 0$ Eq. (4) has $\kappa$ linearly independent solutions. Thus, we obtain $|\kappa|$ Majorana edge states in total.

Now we briefly touch upon the case when a ring $R$ in the complex plane where $Q(z)$ is regular does not exist. This happens, e.g., when the coefficients $t_l$ and $\Delta_l$ exhibit a power-law decay with increasing $l$. If the series (6) for $Q(z)$ is absolutely convergent on the unit circle, then we may still act as described above [20]. If the series is not absolutely convergent, we redefine $Q(z)$ as follows:

$$Q(z) = \sum_{l=-\infty}^{+\infty} (t_l + \Delta_l) z^l e^{-\eta_+ |l|}, \tag{20}$$

where $\eta_+$ is positive. Then, all calculations following Eq. (6) can be reproduced. In the end, one has to go to the limit $\eta_+ \to 0$ to obtain the solutions of Eqs. (3) and (4). Of course, such extension of the Wiener-Hopf technique in not strictly justified, however, it seems reasonable as long as a limit for the Cauchy index exists when $\eta_+$ tends to zero.

We proceed with applications of the developed technique. In the simplest case of chain with finite-range couplings $Q(z)$ is a polynomial, and the Wiener-Hopf technique reproduces the standard solution for $s_l$ in the form of a finite sum of decaying exponents [20]. In the following sections we will consider more complicated model Kitaev chains.

## 3 Power-law model

### 3.1 Power-law falloff of hopping and pairing amplitudes with exponents larger than 1

In this Section, we will analyze the model studied in Refs. [13, 15, 18–20] with power-law falloff of pairing and hopping amplitudes:

$$t_l = \begin{cases} -\mu, & \text{for } l = 0, \\ -\frac{J}{|l|^\alpha}, & \text{for } l \neq 0, \end{cases} \tag{21}$$

$$\Delta_l = \begin{cases} 0, & \text{for } l = 0, \\ \frac{\Delta}{|l|^\beta} \mathrm{sgn}(l), & \text{for } l \neq 0, \end{cases} \tag{22}$$

where $\mu$, $\Delta$ and $J$ are constants. Note that Eqs. (21) and (22) formally encompass the cases of short-range hopping ($\alpha = \infty$) and pairing ($\beta = \infty$).

For a start, we confine ourselves to the case $\alpha > 1$ and $\beta > 1$, so that we can use Eq. (6) to define $Q(z)$ on the unit circle:

$$Q(z) = -\mu + \Delta \left[ \mathrm{Li}_\beta(z) - \mathrm{Li}_\beta(z^{-1}) \right] - J \left[ \mathrm{Li}_\alpha(z) + \mathrm{Li}_\alpha(z^{-1}) \right], \tag{23}$$

where $\mathrm{Li}_\gamma(z)$ is the polylogarithm:

$$\mathrm{Li}_\gamma(z) = \sum_{n=1}^{\infty} \frac{z^n}{n^\gamma}. \tag{24}$$

It can can be demonstrated that the spectrum of an infinite chain is gapped as long as $Q(1) \neq 0$ and $Q(-1) \neq 0$, and that the Cauchy index equals (see Appendix A)

$$\kappa = \begin{cases} 0, & \text{when } Q(1)Q(-1) > 0, \\ 1, & \text{when } Q(1)\Delta > 0 \text{ and } Q(-1)\Delta < 0, \\ -1, & \text{when } Q(-1)\Delta > 0 \text{ and } Q(1)\Delta < 0. \end{cases} \tag{25}$$

Let us consider the special case when $\Delta = J$ and $\alpha = \beta$, so that $Q(z)$ takes the form

$$Q(z) = -\mu - 2J \mathrm{Li}_\alpha(z^{-1}). \tag{26}$$

This function is regular for $|z| > 1$ (including $z = \infty$), and hence $\kappa \leq 0$. We obtain $\kappa = -1$ when (compare with Ref. [15])

$$(\mu + 2J\zeta(\alpha))[\mu - 2J\zeta(\alpha)(1 - 2^{-\alpha})] < 0, \tag{27}$$

where $\zeta(x)$ is the Riemann zeta function. Then, $Q(z)$ has one real root $z = z_0$ in the region $|z| > 1$. If we substitute $Q_+(z) = Q_-^{-1}(z)Q(z)z$ in Eq. (19), we may find that the integral in this equation is determined by the residue of the integrand at $z = z_0$. This yields a solution for $s_l$ of the form $s_l = z_0^{-l}$ – it can be easily checked that this satisfies Eq. (3). If we take $\mu = 0$, we obtain $z_0 = \infty$, which results in a Majorana edge mode localized at the site $l = 0$ [19].

Now let us assume that either $\alpha \neq \beta$, or $\Delta \neq J$, and $\kappa = -1$. The asymptotic behavior of $s_l$ in the limit of large $l$ in this case was determined analytically in Ref. [20], where a connection between the singularities of $Q(z)$ on the unit circle and the behavior of $s_l$ within power-law models with $\alpha > 1$ and $\beta > 1$ has been established using manipulations with Laurent series. A somewhat simpler derivation for our particular model can be found in Appendix A:

$$s_l \propto l^{-\delta}, \qquad \delta \equiv \min(\alpha, \beta). \tag{28}$$

This behavior is also consistent with the one reported in Ref. [19].[1]

## 3.2 Special case of inverse-distance falloff of pairing and hopping amplitudes

The polylogarithm of order 1 has the simple form

$$\mathrm{Li}_1(z) = -\ln(1 - z). \tag{29}$$

Hence, if one considers pairing and hopping amplitudes with asymptotic behavior $t_l, \Delta_l \propto |l|^{-1}$, the function $Q(z)$ acquires logarithmic singularities on the unit circle. Particular examples of this have been analyzed by Pientka *et al.* [5, 16] and Bespalov [7], where pairing and hopping amplitudes were considered that exhibit oscillations together with a $|l|^{-1}$ falloff.

We will briefly review the results obtained for the more general model studied by Bespalov. Within this model, the function $Q(z)$ has the form

$$Q(z) = \epsilon - (\rho + 1)g^*\left[\mathrm{Li}_1(e^{i\varphi_1}z) + \mathrm{Li}_1(e^{i\varphi_2}z^{-1})\right] + (1 - \rho)g\left[\mathrm{Li}_1(e^{i\varphi_2}z) + \mathrm{Li}_1(e^{i\varphi_1}z^{-1})\right] \tag{30}$$
$$+ (1 - \rho)g^*\left[\mathrm{Li}_1(e^{-i\varphi_2}z) + \mathrm{Li}_1(e^{-i\varphi_1}z^{-1})\right] - (\rho + 1)g\left[\mathrm{Li}_1(e^{-i\varphi_1}z) + \mathrm{Li}_1(e^{-i\varphi_2}z^{-1})\right],$$

where $\epsilon$, $\rho$, $\varphi_1$ and $\varphi_2$ are real numbers, $0 < \rho \leq 1$, and $g$ is a complex number. Discrete BdG Hamiltonians characterized by such functions $Q(z)$ arise in the studies of quasiparticle excitations induced by helical magnetic atom chains in bulk superconductors ($\rho = 1$) and constriction-type Josephson junctions ($\rho \leq 1$). The factorization of $Q(z)$ given by Eq. (30) is quite nontrivial. In particular, to determine the behavior of $Q_+(z)$ in the vicinity of its singularities, the right-hand side of Eq. (13) must be evaluated. This has been done in Ref. [7]

---

[1]At the same time, the analytical solution from Ref. [19] of the BdG equations contains mistakes. In particular, the Hamiltonian $\hat{H}_0$ and the projector $\hat{\mathcal{Q}}$ defined in the mentioned paper satisfy $\hat{\mathcal{Q}}\hat{H}_0 = \hat{H}_0$ by definition. The second equation on page 4 of Ref. [19] contradicts this identity. Also note that the final answer for the MZM wave function depends on the arbitrarily chosen Hamiltonian $\hat{H}_0$, and it does not agree with the results from the present paper and Ref. [20]. Still, the asymptotic behavior of the wave function given in Ref. [19] agrees with Eq. (28).

In Ref. [13] the authors try to tackle the problem using the scattering approach. They seek the solution of the BdG equations in the form of a sum of terms proportional to $e^{-\lambda l}$ with $\mathrm{Re}\lambda > 0$, where each term satisfies the BdG equations for an infinite chain. However, for our power-law model and for an infinite chain ($l = -\infty..+\infty$) such exponential solution do not exist, because the series in the right-hand side of Eq. (2) diverge if one substitutes $u_l, v_l \propto e^{-\lambda l}$. This leads to certain difficulties when analyzing this model using the scattering approach. In particular, this approach does not capture the power-law behavior of $s_l$ for $\alpha > 3$ and $\beta > 2$.

in the case when none of the four singular points $e^{i\varphi_1}$, $e^{-i\varphi_1}$, $e^{i\varphi_2}$ and $e^{-i\varphi_2}$ coincides with the others. It was found that when the index $\kappa$ equals $-1$, the asymptotic behavior of $s_l$ in the limit of large $l$ is given by the formula

$$s_l \approx \frac{\cos(l\varphi_1 + \beta_1)}{l(\ln l)^{\frac{3}{2} + \frac{1}{2\rho}}} + \frac{C\cos(l\varphi_2 + \beta_2)}{l(\ln l)^{\frac{3}{2} - \frac{1}{2\rho}}}, \tag{31}$$

where $\beta_1$, $\beta_2$ and $C$ are real numbers that do not depend on $l$. For $\rho = 1$ the number $C$ vanishes, and we obtain $s_l \propto 1/(l\ln^2 l)$ [5, 16].

### 3.3 Power-law falloff of pairing and hopping amplitudes with exponent smaller than 1

Let us now consider the power-law model with either $\alpha < 1$ or $\beta < 1$ (or both). Such model is of fundamental interest due to the unusual logarithmic scaling of the half-chain entanglement entropy with chain length in both the topological and trivial phases [17, 18, 27, 28] (however, since the unusual logarithmic behavior occurs even in the trivial phase, the Majorana mode is not directly related to the scaling of the entanglement entropy, and the Wiener-Hopf technique does not allow to calculate this quantity). In addition, magnetic atom chains on a two-dimensional superconductors, presumably, can be characterized by an effective BdG Hamiltonian with $\alpha = \beta = 1/2$.

As discussed in the literature, there is an ambiguity in the definition of the winding number when $\beta \leq \alpha$ and $\beta < 1$. It has been claimed that the winding number takes half-integer values, which is accompanied by the appearance of so-called massive Dirac edge states, which have a non-vanishing energy for any length of the Kitaev chain [15, 17, 18, 29, 30]. It has been even argued that in this case the ten-fold way classification of topological insulators and superconductors is not applicable, and that the bulk-boundary correspondence is weakened [28].

Let us consider this situation from the point of view of the Wiener-Hopf technique. When $\alpha < 1$ or $\beta < 1$, the function $Q(z)$ is discontinuous on the unit circle at $z = 1$, and we have to use Eq. (20) with $\eta_+ \to +0$ to construct a factorization given by Eq. (11). Let us assume that $\beta < \alpha$. Then, $Q(z)$ has a real root lying in the interval $(e^{-\eta_+}, e^{\eta_+})$. This root tends to 1 in the limit $\eta_+ \to 0$, which makes the winding number ill-defined in this limit. This corresponds to the "half-integer winding number" mentioned above. Then, the edge states have a non-vanishing energy even in long chains, and the developed above formalism does not allow to determine their wave functions.

The situation is different when $\alpha < \beta$. Then, the winding number is well-defined in the limit $\eta_+ \to 0$, and it can be calculated as described in Sec. 3.1, assuming that $Q(1) = -\infty \cdot \text{sgn}(J)$. The analysis of the spatial structure of MZMs given in Ref. [20] is not applicable for $\alpha < 1$, however, we can calculate the MZM wave functions using the explicit formula (13) for the Wiener-Hopf factor. As demonstrated in Appendix A, in the limit of large $l$ we again obtain a power-law falloff of the MZM wave function:

$$s_l \propto l^{(\alpha-3)/2}. \tag{32}$$

In a similar way, the asymptotic behavior of $s_l$ can be determined for $\alpha = \beta < 1$ and for $|J| > |\Delta|$: $s_l \propto l^{-1-\chi}$, where

$$\chi = \pi^{-1}\text{arccot}\left(\frac{J - \Delta - (\Delta + J)\cos(\pi\alpha)}{(J + \Delta)\sin(\pi\alpha)}\right). \tag{33}$$

The quantity $\chi$ lies in the interval $((1-\alpha)/2, 1-\alpha)$ (we assume $J\Delta > 0$, so that $\kappa = -1$). For $|\Delta| > |J|$, the winding number is ill-defined, and the Wiener-Hopf technique is not applicable.

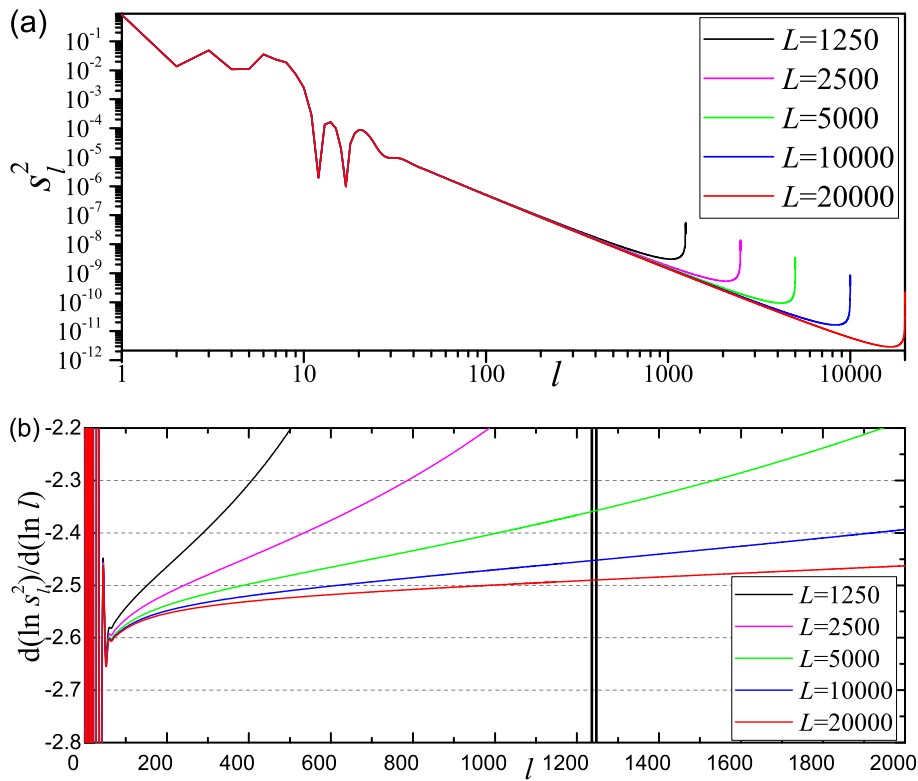

Figure 1: The $s_l^2$ vs. $l$ dependencies (a) and local slope vs. $l$ dependencies (b) for the edge state in a chains with different lengths $L$. The fixed parameters are $\alpha = 0.5$, $\beta = 5.5$, $J = \Delta = 1$, $\mu = 0$.

For an illustration and verification of the analytical results, numerical calculations of the MZM wave functions have been performed for finite chains with up to 20000 sites. Technically, the BdG Hamiltonian $\mathcal{H}$ defined by Eq. (2) was squared, and then the smallest eigenvalue and the corresponding eigenvector were determined. The operator $\mathcal{H}^2$ has a complete system of eigenvectors of the form $(U_l, U_l)^T$ and $(U_l', -U_l')^T$. Thus, the original eigenvalue problem in a $2L$-dimensional space was reduced to two similar problem in $L$-dimensional subspaces. The eigenvectors $(U_l, U_l)^T$ of $\mathcal{H}^2$ are connected with the eigenvectors $(u_l, v_l)^T$ of $\mathcal{H}$ via $U_l = (u_l + v_l)/\sqrt{2}$. This means that for $\kappa = -1$ the coefficients $U_l$ corresponding to the smallest eigenvalue of $\mathcal{H}^2$ are approximately proportional to $s_l$ for the MZM. Several profiles of $s_l^2 = (u_l + v_l)^2$ corresponding to the solution of Eq. (2) with the smallest positive energy are shown in Fig. 1a in double logarithmic scale. Different curves correspond to different chain lengths $L$. Curves corresponding to length $L$ and $2L$ roughly coincide for $l \leq L/3$ – the convergence of $s_l$ with increasing $L$ is relatively slow (the situation is drastically different for large $\alpha$ and $\beta$ – see below). The curve corresponding to $L = 20000$ has a large segment with seemingly linear behavior, however, in fact, the slope of the curve changes significantly in this segment: for $l = 1000..2000$ a linear least-square fit yields a slope of $-2.48$, while for $l = 4000..5000$ the slope is $-2.36$. To illustrate this feature more clearly, let us define the local slope as

$$\frac{\mathrm{d} \ln s_l^2}{\mathrm{d} \ln l} \equiv \ln(s_{l+1}^2/s_l^2) l \,. \tag{34}$$

This quantity should approach $\alpha - 3$ for large $l$ and $\alpha < 1$, according to Eq. (32). In Fig. 1b, one can see how the local slope scales with chain length. Here, convergence is even slower than for $s_l^2$: the curves corresponding to $L = 10000$ and $L = 20000$ diverge noticeably for

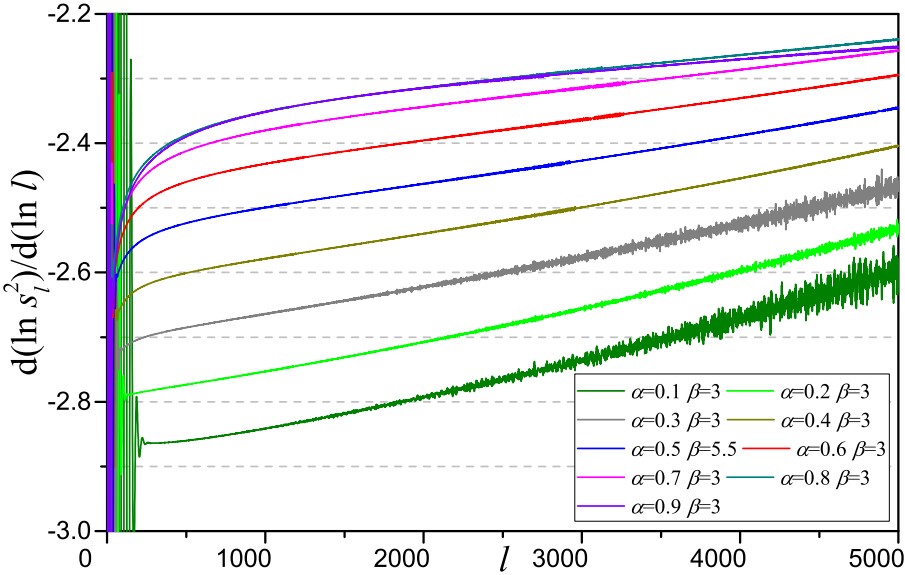

Figure 2: Local slope of the $\ln s_l^2$ vs. $\ln l$ dependence [Eq. (34)] for chains with length $L = 20000$ and different parameters $\alpha$ and $\beta$. Other parameters are $J = \Delta = 1$, $\mu = 0$.

$l > 200$. We may argue that the local slope for $L = 20000$ corresponds to the local slope of an isolated MZM for $l \lesssim 200$ – presumably, this is insufficient to reach the limit of large $l$, where Eq. (32) should work well. Thus, finite-size effects are very strong for $\alpha < 1$, when the hopping amplitudes $t_l$ fall off very slowly with distance between sites.

Some more profiles of the local slope, corresponding to different parameters $\alpha < 1$ and $\beta$, in the vicinity of the left end of a the chain with length $L = 20000$ are shown in Fig. 2b. The slopes exhibit significant variations with site number $l$, which is most likely due to finite-size effects. Still, some features predicted for semi-infinite chains can be seen: the local slope lies between $-2$ and $-3$, and for $0.1 \leq \alpha \leq 0.8$ it increases with $\alpha$, in accordance with Eq. (32). The curves seem to clump up as $\alpha$ approaches 1. This anomaly is connected with the breakdown of the power-law behavior of $s_l$ for $\alpha$ close to 1, as discussed in Sec. 3.2.

For comparison, in Fig. 3 profiles of $s_l^2$ and of the local slope for chains with different lengths and parameters $\alpha = 6$ and $\beta = 4$ are shown. Unlike in the case $\alpha < 1$, now the curves corresponding to different $L$ almost ideally coincide, with significant differences only in the vicinity of the right edge of the shorter chain. Even the "noisy" features on the graph for the local slope match perfectly for different $L$ (these features remained practically unchanged when a different algorithm for the solution of the eigenvalue problem was chosen; their origin is yet to be understood). A linear least-square fit of the curve corresponding to $L = 20000$ in Fig. 3a yields a slope equal to -8.013 in the range $l = 1000..2000$ and a slope equal to -8.003 in the range $l = 4000..5000$ – variations of the slope are small compared to the case $\alpha < 1$. The edge mode wave function clearly exhibits a power law falloff, in good agreement with Eq. (28) (a power law falloff for $\alpha > 1$ and $\beta > 1$ has been also found numerically in Refs. [18, 19]).

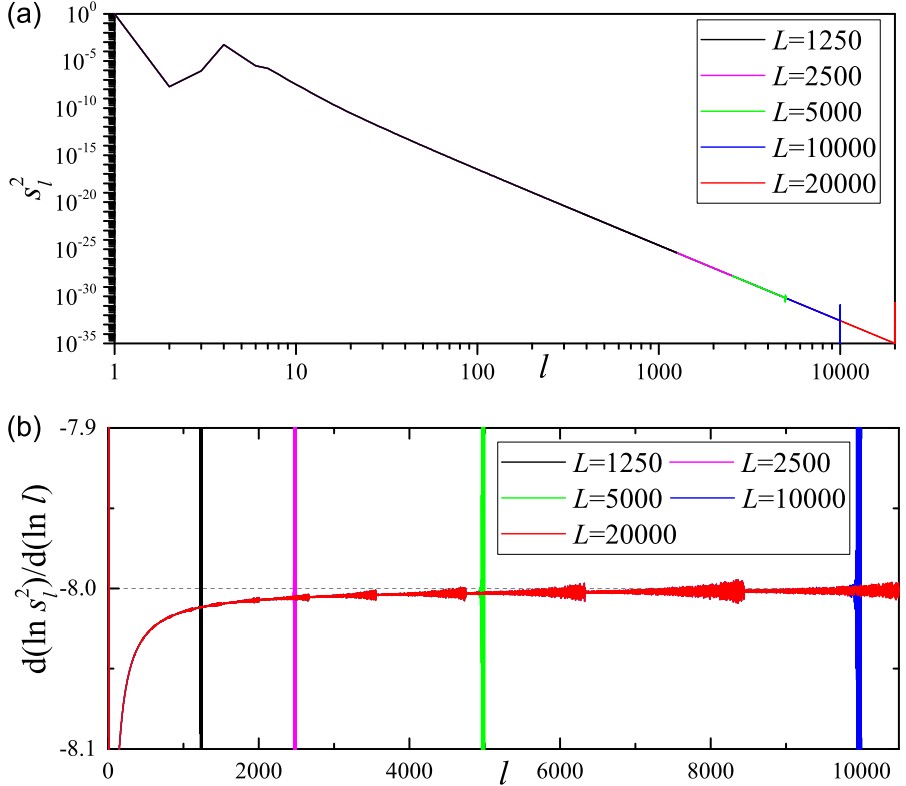

Figure 3: The $s_l^2$ vs. $l$ dependencies (a) and local slope vs. $l$ dependencies (b) for the edge state in a chains with different lengths $L$. The fixed parameters are $\alpha = 6$, $\beta = 4$, $J = \Delta = 1$, $\mu = 0$. Curves corresponding to different $L$ overlap almost perfectly (except for cites close to the right end of the chain).

## 4 Exponential falloff of hopping amplitudes

In Ref. [29], the boundaries of the topological phases for the following model with short-range pairing and long-range hopping with exponential falloff with distance have been discussed:

$$t_l = \begin{cases} -\mu, & \text{for } l = 0, \\ t_1 a^{|l|-1}, & \text{for } l \neq 0, \end{cases} \tag{35}$$

$$\Delta_l = \begin{cases} 0, & \text{for } l \neq \pm 1, \\ \Delta \operatorname{sgn}(l), & \text{for } l = \pm 1, \end{cases} \tag{36}$$

where $0 < a < 1$. The function $Q(z)$ equals

$$Q(z) = \frac{t_1 z}{1 - az} + \frac{t_1}{z - a} - \mu + \Delta(z - z^{-1}). \tag{37}$$

To determine its Cauchy index, the same considerations as in Appendix A can be used, which yield that the Cauchy index is given by the same equation as for the power-law model – see Eq. (25).

Let us consider the case $\kappa = -1$, when Eq. (3) has a nontrivial solution. The function $Q(z)$ is rational, so that the factors $Q_+(z)$ and $Q_-(z)$ can be found even without using Eqs. (13) and (14) if the poles and roots of $Q(z) = 0$ are known. The equation $Q(z) = 0$ is equivalent to a quartic equation, and its four roots can be calculated analytically if desired.[2] When $\kappa = -1$,

---

[2]https://en.wikipedia.org/wiki/Quartic_equation.

three of these roots – $z_1$, $z_2$ and $z_3$ – lie outside the unit circle. These roots as well as the pole at $z = a^{-1}$ are contained in the factor $Q_+(z)$:

$$Q_+(z) = \frac{(z - z_1)(z - z_2)(z - z_3)}{1 - az}. \tag{38}$$

The integral in Eq. (3) is determined by the residues at the three poles of $Q_+^{-1}(z)$:

$$s_l = -\sum_{n=1}^{3} \frac{1}{z_n^{l+1} Q_+'(z_n)}. \tag{39}$$

The asymptotic behavior of $s_l$ in the limit $l \to \infty$ is determined by the root (or roots) of $Q_+(z)$ with the smallest absolute value. Let us derive some properties of these roots. One of these roots is always negative, because $Q(-1)$ and $Q(-\infty)$ have different signs. The other two roots can be real or complex. In the limit of very large $|\Delta|$, an approximate solution of the equation $Q(z) = 0$ yields

$$z_1 \approx -1 - \frac{Q(-1)}{2\Delta}, \qquad z_2 \approx 1 - \frac{Q(1)}{2\Delta}, \tag{40}$$

$$z_3 \approx a^{-1} + \frac{t_1 a^{-1}}{\Delta(1 - a^2)}. \tag{41}$$

In the opposite limit of small $|\Delta|$ we can obtain

$$z_1 \approx \frac{\mu + a^{-1} t_1}{\Delta}, \tag{42}$$

$$z_2 \approx z_2^{(0)} \left( 1 - \frac{\Delta \left| 1 - a z_2^{(0)} \right|^2}{t_1 + a\mu} \right), \qquad z_3 = z_2^*, \tag{43}$$

$$z_2^{(0)} = \frac{2 t_1 a + (a^2 + 1)\mu + i(1 - a^2)\sqrt{-Q(1)Q(-1)}}{2(t_1 + a\mu)}. \tag{44}$$

## 5  Energy of the edge mode in a finite chain

In this Section, we will obtain estimates for the energy of the edge states composed of two MZMs in finite chains. We will consider the Hamiltonian (1) with $l, n = 0..L$. For simplicity, we assume that the Cauchy index of $Q(z)$ equals $-1$. Then, there is one MZM corresponding to the left edge and one mode corresponding to the right edge of the chain (a generalization for multiple MZMs per edge is discussed in Ref. [20]). Their wave functions are $(u_l, u_l)^T$ and $(u_{L-l}, -u_{L-l})^T$, respectively. The coefficient $u_l$ satisfy the equations

$$\sum_{n=0}^{\infty} (t_{l-n} + \Delta_{l-n}) u_n = 0, \quad l \geq 0. \tag{45}$$

The eigenfunction of the BdG Hamiltonian is a superposition of the two defined above wave functions with equal weights. The energy $E_0$ corresponding to this eigenfunction can be estimated as the matrix element of the BdG Hamiltonian between $(u_l, u_l)^T$ and $(u_{L-l}, -u_{L-l})^T$:

$$E_0 \sim \sum_{n,l=0}^{L} (u_{L-l}, -u_{L-l}) \begin{pmatrix} t_{l-n} & \Delta_{l-n} \\ \Delta_{n-l}^* & -t_{n-l} \end{pmatrix} \begin{pmatrix} u_n \\ u_n \end{pmatrix}$$

$$= 2 \sum_{n,l=0}^{L} u_{L-l}(t_{l-n} + \Delta_{l-n}) u_n. \tag{46}$$

For further transformations we use Eq. (45):

$$E_0 \sim -2 \sum_{l=-\infty}^{-1} \sum_{n=0}^{L} u_{L-l}(t_{l-n} + \Delta_{l-n})u_n . \tag{47}$$

To obtain an order of magnitude estimate for $E_0$, we need to know the coefficients $u_l$ with appropriate normalization, which we did not calculate. However, to determine the asymptotic behavior of $E_0$ in the limit of large $L$, we need to know only the behavior of $u_l$ in the limit of large $l$.

First, let us consider the case when the MZM wave function falls off exponentially fast in the limit of large $l$: $u_l \propto e^{-bl}$, $b > 0$. We have then

$$E_0 \propto \sum_{l=-\infty}^{-1} \sum_{n=0}^{L} e^{bl-bL}(t_{l-n} + \Delta_{l-n})u_n$$

$$\approx e^{-bL} \sum_{l=-\infty}^{-1} \sum_{n=0}^{\infty} e^{bl}(t_{l-n} + \Delta_{l-n})u_n \propto e^{-bL} . \tag{48}$$

The double series here is always convergent: the sum is proportional to $p(e^b)$ [see Eq. (8)].

Next, consider the case when $t_l$ and $\Delta_l$ are given by Eqs. (21) and (22), respectively, and $\delta \equiv \min(\alpha, \beta) > 1$. Then, the asymptotic behavior of the MZM wave functions is given by Eq. (28). We have then in the limit of large $L$

$$E_0 \propto \sum_{l=-\infty}^{-1} \sum_{n=0}^{L} \frac{1}{|L-l|^\delta}(t_{l-n} + \Delta_{l-n})u_n$$

$$\approx \frac{1}{L^\delta} \sum_{l=-\infty}^{-1} \sum_{n=0}^{\infty} (t_{l-n} + \Delta_{l-n})u_n \propto L^{-\delta} . \tag{49}$$

The series here is convergent as long as $\delta > 1$. Equation (49) is consistent with the numerical $E_0$ vs. $L$ dependence found in Ref. [19] and with the analytical result from Ref. [20].

When $\delta < 1$ and $\alpha < \beta$, we have $E_0 \propto L^{-1}$, as demonstrated in Appendix B. Some $E_0$ vs. $L$ dependences, calculated by solving numerically the BdG equations as described in Sec. 3.3, are shown in Fig. 4 in double logarithmic scale. Unlike the edge mode wave function, the energy exhibits power-law behavior already for $L \geq 40$: $E \propto L^k$ with $k$ very close to $-1$ (except for the case when $\alpha$ is very close to 1), in good agreement with the analytical result.

## 6 Conclusion

To sum up, we have applied the Wiener-Hopf technique to analyze the Majorana edge modes in Kitaev chains belonging to the BDI symmetry class with infinite-range couplings between sites. Using explicit formulas for the Wiener-Hopf factors, we developed an efficient analytical method to calculate the energies of Majorana modes in long open chains and asymptotic behaviors of MZM wave functions at large distances $l$ from the edge of a semi-infinite chain. In particular, we considered a model chain with exponential falloff of hopping amplitudes, where the MZM wave functions fall off exponentially with $l$, and the energy $E_0$ of the edge mode in a finite chain also decays exponentially with chain length $L$.

The paper is mainly focused on a model with power-law falloff of hopping and pairing amplitudes: $t_n \propto |n|^{-\alpha}$, $\Delta_n \propto |n|^{-\beta}$. For $\alpha > 1$ and $\beta > 1$ we reproduce analytical and numerical results from literature. For $\alpha < \beta$ and $\alpha < 1$, we find that the MZM wave function falls off as $l^{(\alpha-3)/2}$, and for $\alpha = \beta < 1$ the wave function exhibits a power-law falloff with a

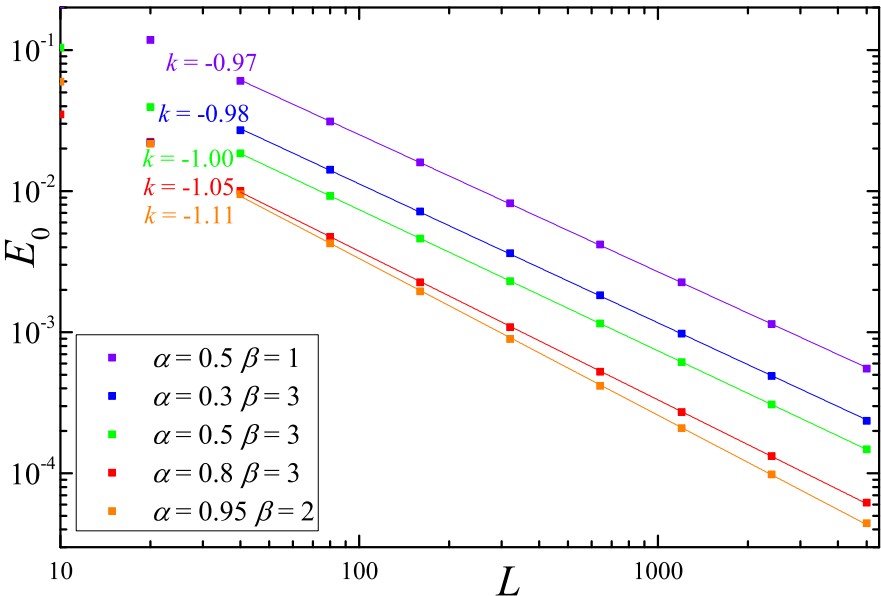

Figure 4: Edge mode energy $E_0$ vs. chain length $L$ dependences for the power-law model [Eqs. (21) and (22)] with different parameters $\alpha$ and $\beta$. Other parameters are $J = \Delta = 1$, $\mu = 0$. The lines represent linear least-square fits for points with $L \geq 40$. The slopes $k$ of the lines are indicated next to them.

more complicated exponent. Remarkably, for $\alpha < \beta$ and $\alpha < 1$ the decay rate of the MZM wave function decreases with increasing $\alpha$ (as opposed to the case $1 < \alpha < \beta$). This feature can be observed in finite open chains with length $L \gtrsim 10000$, as has been demonstrated by solving numerically the BdG equations. However, the asymptotic behavior of the edge mode does not quite follow a power law even in chains with length $L = 20000$. The analytically predicted behavior of the energy in the limit of large $L$ is $E_0 \propto L^{-1}$ for $\alpha < 1$, $\alpha < \beta$. This behavior can be observed even in short chains with length $L \geq 40$.

The obtained results are relevant for the description of Majorana modes in magnetic atom chains on superconductors (which can be studied using scanning tunneling microscopy) and in chains of cold atoms in optical lattices.

# Acknowledgments

The author is grateful to A.S. Mel'nikov for useful comments on the paper.

**Funding information** The work has been supported by Russian Science Foundation grant No. 21-12-00409.

# A  Analysis of the model with power-law falloff of pairing and hopping amplitudes

This appendix contains some technical details related to the analysis of MZM wave functions within the power-law model defined by Eqs. (21) and (22).

First, we will determine the value of the Cauchy index of the function $Q(z)$ given by Eq. (23). We will use the integral representation of the polylogarithm [31]:

$$\text{Li}_\gamma(z) = \frac{z}{\Gamma(\gamma)} \int_0^\infty \frac{x^{\gamma-1} dx}{e^x - z}, \tag{A.1}$$

where $\Gamma(\gamma)$ is the gamma function. For the imaginary part of $Q(z)$ on the unit circle we obtain

$$\begin{aligned} \text{Im}[Q(e^{i\varphi})] &= 2\Delta \text{Im}[\text{Li}_\beta(e^{i\varphi})] \\ &= \frac{2\Delta \sin\varphi}{\Gamma(\beta)} \int_0^\infty \frac{e^x x^{\beta-1} dx}{|e^x - e^{i\varphi}|^2}. \end{aligned} \tag{A.2}$$

The right-hand side here is non-zero for $\varphi \neq 0$ and $\varphi \neq \pi$, which indicates that the bulk spectrum of the chain is gapped, unless $Q(1) = 0$ or $Q(-1) = 0$. It also follows from this that the modulus of the winding number $\kappa$ can not exceed unity, so that $\kappa$ takes the values -1, 0 or 1. In the general case, when $Q(1)Q(-1) > 1$ the Cauchy index is even because $Q(z^*) = [Q(z)]^*$, and hence $\kappa = 0$ within our power-law model. The Cauchy index is odd when $Q(1)Q(-1) < 1$, and its sign coincides with the sign of $Q(1)\Delta$.

Now we will determine the asymptotic behavior of $s_l$ in the limit of large $l$ when either $\alpha \neq \beta$, or $\Delta \neq J$. Then, outside the unit circle $Q(z)$ has branch points at $z = 1$ and $z = \infty$, which should be connected by a branch cut. In Eq. (19) we can integrate along the contour $\ell$ shown in Fig. 5 instead of the unit circle. Such transformation is allowed if $Q(z)$ has no zeros between the unit circle and the contour $\ell$. This contour consists of two segments: $\ell_1$ goes along the circle $|z| = r$, and $\ell_2$ encloses the branch cut. Correspondingly, we can break down $s_l$ into two terms:

$$s_l = s_l^{(1)} + s_l^{(2)}, \tag{A.3}$$

where

$$s_l^{(1)} = \frac{1}{2\pi i} \int_{\ell_1} \frac{dz}{Q_+(z)z^{l+1}} = \frac{1}{2\pi r^l} \int_0^{2\pi} \frac{e^{-il\beta} d\beta}{Q_+(re^{il\beta})}, \tag{A.4}$$

$$s_l^{(2)} = \frac{1}{2\pi i} \int_{\ell_2} \frac{dz}{Q_+(z)z^{l+1}}. \tag{A.5}$$

The integral in the right-hand side of Eq. (A.4) is bounded, hence, $s_l^{(1)}$ is exponentially small in the limit of large $l$.

To transform $s_l^{(2)}$, we substitute $Q_+(z) = Q(z)Q_-^{-1}(z)z$ into Eq. (A.5) and switch to the integration variable $t = l \ln z$:

$$s_l^{(2)} = \frac{1}{2\pi i l} \int_0^{l\ln r} \frac{Q_-(e^{t/l})}{e^t} \left[ \frac{1}{Q(e^{t/l+i0})} - \frac{1}{Q(e^{t/l-i0})} \right] dt. \tag{A.6}$$

To determine $Q(z)$ on both sides of the branch cut, we use the relation

$$\text{Li}_\gamma(e^y) = \Gamma(1-\gamma)(-y)^{\gamma-1} + \sum_{n=0}^\infty \zeta(\gamma-n)\frac{y^n}{n!}, \tag{A.7}$$

which is applicable for $|y| < 2\pi$ and for non-integer $\gamma$ [31]. For integer $\gamma$ [32]

$$\text{Li}_\gamma(e^y) = \left[ \sum_{n=1}^{\gamma-1} \frac{1}{n} - \ln(-y) \right] \frac{y^{\gamma-1}}{(\gamma-1)!} + \sum_{n=0, n \neq \gamma-1}^\infty \zeta(\gamma-n)\frac{y^n}{n!}. \tag{A.8}$$

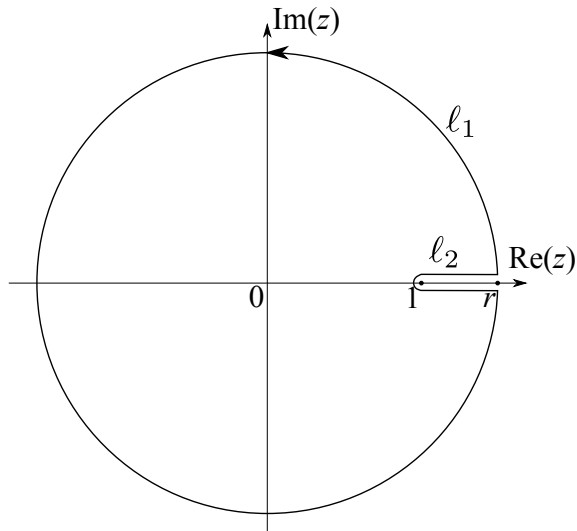

Figure 5: Deformed integration contour $\ell$.

In both cases, we have

$$\mathrm{Li}_\gamma(e^{y+i0}) - \mathrm{Li}_\gamma(e^{y-i0}) = \frac{2\pi i}{\Gamma(\gamma)} y^{\gamma-1}, \tag{A.9}$$

for $y > 0$, and

$$Q(e^{y+i0}) - Q(e^{y-i0}) = \frac{2\pi i \Delta}{\Gamma(\beta)} y^{\beta-1} - \frac{2\pi i J}{\Gamma(\alpha)} y^{\alpha-1}. \tag{A.10}$$

In the limit of large $l$ we transform $s_l^{(2)}$ as follows:

$$
\begin{aligned}
s_l^{(2)} &= \int_0^{l\ln r} \frac{e^{-t} Q_-(e^{t/l})}{|Q(e^{t/l+i0})|^2} \left[ \frac{\Delta}{\Gamma(\beta)} \frac{t^{\beta-1}}{l^\beta} - \frac{J}{\Gamma(\alpha)} \frac{t^{\alpha-1}}{l^\alpha} \right] dt \\
&\approx \frac{Q_-(1)}{Q(1)^2} \int_0^\infty \left[ \frac{\Delta}{\Gamma(\beta)} \frac{t^{\beta-1}}{l^\beta} - \frac{J}{\Gamma(\alpha)} \frac{t^{\alpha-1}}{l^\alpha} \right] e^{-t} dt = \frac{Q_-(1)}{Q(1)^2} \left[ \frac{\Delta}{l^\beta} - \frac{J}{l^\alpha} \right].
\end{aligned}
\tag{A.11}
$$

Strictly speaking, only the leading term should be kept in the right-hand side: if we take into account corrections to $s_l^{(2)}$ due to the weak dependence of $Q_-(e^{t/l})$ and $Q(e^{t/l+i0})$ on $t$, we obtain terms proportional to $l^{-\alpha-\gamma}$ and $l^{-\beta-\gamma}$ in the asymptotic expansion of $s_l^{(2)}$ with some positive numbers $\gamma$. Thus, we arrive at Eq. (28).

Similar calculations can be performed in the case $\alpha < 1$, $\alpha < \beta$. First, we determine $Q_+(z)$ for $z$ close to 1 by integrating in Eq. (13) along the deformed contour shown in Fig. 5:

$$\frac{Q_+(z)}{Q_1(z)} = \exp\left( \frac{1}{2\pi i} \int_1^r \frac{\ln Q(t+i0) - \ln Q(t-i0)}{t-z} dt \right), \tag{A.12}$$

$$Q_1(z) = \exp\left( \frac{1}{2\pi i} \int_{\ell_1} \frac{\ln(Q(t) t^{-\kappa})}{t-z} dt \right). \tag{A.13}$$

The logarithm in Eq. (A.12) should be continuous on the contour $\ell_2$ (for $\eta_+ > 0$). Using Eqs. (23) and (A.7), on the real axis for $z > 1$ and in the vicinity of $z = 1$ in the leading order we obtain

$$Q(z \pm i0) \approx -2J\Gamma(1-\alpha) \sin\left( \frac{\pi\alpha}{2} \right) e^{\pm i\pi(1-\alpha)/2} (z-1)^{\alpha-1}. \tag{A.14}$$

Substituting this into Eq. (A.12) we find that

$$Q_+(z) \approx Q_1(z) \exp\left(\frac{1-\alpha}{2}\int_1^r \frac{dt}{t-z}\right) = \tilde{Q}_1(z)(1-z)^{(\alpha-1)/2}, \tag{A.15}$$

where $\tilde{Q}_1$ is approximately constant in the vicinity of $z = 1$. Next, we substitute Eq. (A.15) into Eq. (19) assuming $\kappa = -1$. Again, we integrate along the deformed contour shown in Fig. 5 to obtain in the limit of large $l$ (see similar calculations above)

$$
\begin{aligned}
s_l &\approx \frac{1}{2\pi i}\int_{\ell_2}\frac{dz}{Q_+(z)z^{l+1}}\\
&\approx -\frac{\tilde{Q}_1^{-1}(1)}{\pi}\cos\left(\frac{\pi\alpha}{2}\right)\int_1^r \frac{dz}{z^{l+1}}(z-1)^{\frac{1-\alpha}{2}}\\
&\approx -\frac{\tilde{Q}_1^{-1}(1)}{\pi l}\cos\left(\frac{\pi\alpha}{2}\right)\int_0^\infty e^{-t}(e^{t/l}-1)^{\frac{1-\alpha}{2}}dt\\
&\approx -\frac{\tilde{Q}_1^{-1}(1)}{\pi l}\cos\left(\frac{\pi\alpha}{2}\right)\int_0^\infty e^{-t}\left(\frac{t}{l}\right)^{\frac{1-\alpha}{2}}dt = -\frac{\tilde{Q}_1^{-1}(1)}{\pi l^{\frac{3-\alpha}{2}}}\cos\left(\frac{\pi\alpha}{2}\right)\Gamma\left(\frac{3-\alpha}{2}\right).
\end{aligned}
\tag{A.16}
$$

# B  Asymptotic behavior of the energy $E_0$ within the power-law model with $\alpha < 1$

In this appendix, we will determine the asymptotic behavior of the energy $E_0$ in the limit of large chain lengths $L$ within the model where $t_l$ and $\Delta_l$ are given by Eqs. (21) and (22), respectively, and additionally $\alpha < 1$, $\alpha < \beta$. We also assume that $\kappa = -1$. For a start, we transform Eq. (47):

$$E_0 \sim S_1 + S_2, \tag{B.1}$$

$$S_1 = -2\sum_{l=-\infty}^{-1}\sum_{n=0}^{\infty}u_{L-l}(t_{l-n}+\Delta_{l-n})u_n, \tag{B.2}$$

$$S_2 = 2\sum_{l=-\infty}^{-1}\sum_{n=L+1}^{\infty}u_{L-l}(t_{l-n}+\Delta_{l-n})u_n. \tag{B.3}$$

It is relatively easy to determine the asymptotic behavior of $S_2$:

$$
\begin{aligned}
S_2 &\propto \sum_{l=-\infty}^{-1}\sum_{n=L+1}^{\infty}\frac{1}{(L-l)^{\frac{3-\alpha}{2}}(l-n)^{\alpha}n^{\frac{3-\alpha}{2}}}\\
&\approx \int_{-\infty}^0 dl\int_L^\infty dn\,\frac{1}{(L-l)^{\frac{3-\alpha}{2}}(l-n)^{\alpha}n^{\frac{3-\alpha}{2}}}\\
&= \frac{1}{L}\int_{-\infty}^0 dl\int_1^\infty dn\,\frac{1}{(1-l)^{\frac{3-\alpha}{2}}(l-n)^{\alpha}n^{\frac{3-\alpha}{2}}}\propto \frac{1}{L}.
\end{aligned}
\tag{B.4}
$$

To estimate $S_1$, we use that fact that $u_n = s_n/2$:

$$S_1 = -\frac{1}{2}\sum_{l=-\infty}^{-1}s_{L-l}p_l, \tag{B.5}$$

where the coefficients $p_l$ are introduced in Eq. (9). The sum in Eq. (B.5) equals an expansion coefficient in the Laurent series of $s(z)p(z)$, so that

$$S_1 = -\frac{1}{4\pi i} \oint_{|z|=1} \frac{p(z)s(z)dz}{z^{L+1}}\,. \tag{B.6}$$

From Eq. (15) we find that $s(z) = Q_+^{-1}(z)$, $p(z) = Q_-(z)/z$ up to a normalization factor, which does not depend on $L$. Taking into account Eq. (11), we obtain

$$S_1 \propto \frac{1}{4\pi i} \oint_{|z|=1} \frac{Q(z)dz}{Q_+^2(z)z^{L+1}}\,. \tag{B.7}$$

Further calculations are made in the same spirit as the calculations of $s_l$ in Appendix A. We integrate along the deformed contour shown in Fig. 5 and neglect the exponentially small contribution from the segment $\ell_1$. Using Eqs. (A.14) and (A.15), we obtain

$$\begin{aligned}
S_1 &\propto \frac{1}{4\pi i} \int_1^r \frac{dz}{z^{L+1}} \left[ \frac{Q(z+i0)}{Q_+^2(z+i0)} - \frac{Q(z-i0)}{Q_+^2(z-i0)} \right] \\
&\propto \int_1^r \frac{dz}{z^{L+1}} \propto \frac{1}{L}\,.
\end{aligned} \tag{B.8}$$

Both components of $E_0 - S_1$ and $S_2$ – are proportional to $L^{-1}$. Strictly speaking, these contributions can cancel each other out, however, numerical diagonalizations of the BdG Hamiltonians of finite chains performed by the author demonstrate that this does not happen, and the energy $E_0$ is proportional to $L^{-1}$.

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
