# Peer review of "Majorana edge states in Kitaev chains of the BDI symmetry class"

_SciPost Physics, doi:SciPost Phys. Core 6, 080 (2023)_

## Round 1 · Referee Report · Anonymous (Referee 1) · 2023-6-13

Strengths

1) Clarity and exhaustiveness of the presentation.

2) Novel and relevant technical developments.

Weaknesses

3) Lacks further discussion of crucial aspect of the problem.

Report

I have read the manuscript by the author and I was positively impressed by the in depth investigation of the edge state of the linear chain with different couplings shape. The outlined analytic method appears to be very powerful and i have no reason to doubt the correctness of the calculations. I am inclined to recommend the paper for publication provided that the author is willing to consider a few revision.

The major question mark concerning the paper appears in below Eq. (33) where the author reproduces the result of Ref. [18], but the author expresses some doubts about whether this is the leading behaviour. The author shall make some major comment on what is expected to occur to Eq. (33) upon including further terms in the expansion in Eq. (A10). Indeed, the recent work [arXiv:2301.12514 ] showed how the result in Eq. (33) is not recovered by the scattering approach for $\alpha,\beta>2$, since the leading order momentum terms of the dispersion relation change across this boundary. The author should also acknowledge Ref. [arXiv:2211.15690] which deals on similar matters.

It would be also interesting to comment on how the edge states influences other quantities aside the energy, such as the entanglement entropy of the open hand portion of the chain with respect to the infinite system. In this perspective, please notice that enhanced entanglement scaling is one of the major features of long-range Kitaev chains

10.1088/1742-5468/ac7644

https://doi.org/10.1007/JHEP05(2023)066

Requested changes

1) Comment on the long distance scaling of edge states for \alpha,\beta>2

2) Comment on the possibility to evaluate the entanglement entropy of the system with the method described in the paper.

---

## Round 1 · Referee Report · Anonymous (Referee 2) · 2023-7-4

Strengths

  1. The paper presents a novel approach to the problem that makes use of robust and rigorous analytical techniques.
  2. The results are presented in a very clear and exhaustive way.

Weaknesses

The paper studies the structure of edge states, but does not address other questions about the description of the different quantum phases, especially in the controversial regime of alpha,beta<1.

Report

The paper deals with a class of interesting models (Kitaev chain with long-range interactions) that, despite their apparent simplicity, displays unusual properties and a rich physics that is yet to be fully understood. The proposed rigorous analytical method is crucial to obtain important results about edge states in this model, which are carefully explained in the manuscript.
The only concern is about a more precise comparison with previous results, such as the one cited in the bibliography (in particular ref. [18]) as well as more recent ones (e.g. Phys. Rev. Lett. 130, 246601 (2022), arXiv:2301.12514). This also implies to enlarge the analysis to other quantities, such as entanglement entropy and/or correlation functions that might shed light on the universal properties of the models. I understand this might be a lot of work, but at least some comments are at order.
The results are interesting and I would suggest publication, after requested changes are taken into account.

Requested changes

Comments about: 1. a stricter comparison with previous results and 2. about the possibility to calculate other physical quantities (e.g. entanglement entropy and/or correlation functions).

---

## Round 2 · Referee Report · Anonymous (Referee 2) · 2023-8-3

Strengths

  1. The analytical results are strong
  2. Good presentation and comparison with previous results

Weaknesses

  1. Numerical checks are not properly explained and interpreted

Report

The authors has carefully considered the Referees' suggestions and changed the paper accordingly.
In particular, now recent literature is correctly mentioned and acknowledged.
I think the theoretical and analytical calculations are now complete.

However I have some questions regarding the added part on numerical evaluations of s_l and E_0(l).
Finite size effects can be very strong (especially when alpha<1) so that it is necessary to consider very long chains and values of l far from the (right) boundary of the finite chain considered in numerics. Also the theoretical predictions holds for l large. i.e. far from the left boundary l=0. This can be clearly seen in Fig 1(a), from which one can reasonably argue that the correct range to perform any fit lies between l=40 and l=5000, at least for L=20000.
This is indeed the range considered for the evaluation of the energy E_0(l) (Fig.2), which exhibits results agreeing with the theoretical predictions. Instead, the author has not used the same procedure for s_l and I wonder why a similar fit (in the same range) is not reported for curves such as the one reported in Fig 1(a), for different values of alpha.
Also, it would be appropriate to have an estimate of the error of the fit.
Finally, I have doubts on the relevance of the behaviour of d(ln s_l^2)/d(ln l) as function of l itself -shown in Fig 1(b). This quantity is calculated locally from a quantity such as s_l that on a chain of length L is for sure affected strongly by both the finiteness of the chain itself and of the lattice spacing.

Requested changes

Analysis of the validity of the numerics

---

## Round 2 · Referee Report · Anonymous (Referee 1) · 2023-8-10

Report

I read the revised manuscript from the author and I acknowledge that he made a serious effort to address mine and the other referee concerns. Unfortunately, part of these modifications are bound to generate confusion rather than clarify the open issues. In my previous report I requested the author to clarify the issue of the decay of the Majorana edge states for power-law interactions with \alpha,\beta>2 due to the mismatch between the result of the author (and of Ref. [20]) with the ones of Ref. [13]. The author, in agreement with Ref.[20], obtains the power-law decay in Eq.(28)

s_{l}\propto l^{-\delta} with \delta=min(\alpha,\beta).

Ref.[13] finds that the above result is only applicable for \alpha<3 or \beta<2, while at larger decays the approximation of Ref.[13] yields that long-range interactions are in-influential to the edge modes decay.

In the revised version of the manuscript the author claims that the result in Eq.(28) is the leading one and that all additional terms are subleading. In order to justify the discrepancy with the finding of Ref.[13] the author also makes some general claims in a footnote (Ref. [29]), where he states that Ref.[13] is "most likely erroneous". In order to sustain this latter claim the author states that assuming $u_{l},v_{l}\propto exp(-\lambda l)$ with $Re(\lambda)>0$ yields a diverging result in Eq.(2) .

Actually, it is the claim of the author to be erroneous. In fact, if one assumes the exponential decay of the BdG amplitudes with positive decay exponent the summation on the l.h.s. of Eq.(2) is convergent and, accordingly, cannot serve as an argument against the scattering approach. I advice the author to remove his fallacious statement and amend the footnote accordingly. Also, I would advise the author to be more cautious when implicating possible mistakes in other published works.

More in general, it should be noted that all approaches of Ref. [13], of Ref.[20] and of the current manuscript, albeit using different methods, produce similar equations. The main discrepancy between the approaches of Ref. [13] and Ref.[20] depends on the way the polylogarithm singularities in the complex plane are treated. Indeed, Ref.[20] always keeps the non-analytic term in the expansion of the PolyLog. On the contrary, Ref.[13] only focuses on the leading order contribution, which becomes analytic for $\alpha>3$ and $\beta>2$.

The author's result is specular to the one of Ref.[20], but the author states that Eq. (28) can only be applied to the case of non-integer $\alpha,\beta$. This is in agreement with the aforementioned picture in which the non-analyticity of the polylog function plays a crucial role in the Wiener-Hopf method. However, this result leaves open two alternative scenarios:

a) For integer power-laws e.g. $\alpha=6$ and $\beta=4$, the decay of the modes is still power-law.

b) The power-law behaviour only occurs at non-integer values and the integer case displays exponential behaviour.

The author stated that his analysis unambiguously identifies Eq. (28) as the leading behaviour for the decay of the edge mode, but what scenario is to be expected? In the case (a) the author shall clarify how the power-law behaviour is recovered in absence of the polylog branch cut. In the case (b) the result will be quite unphysical since all integer power-law decays will behave differently from the non-analytic case. These question is not physically irrelevant since most of the naturally occurring power-law potentials display integer power-laws (gravity, coulomb interaction, dipolar, van-der-Wals,...).

In the revised version of the manuscript the author also includes some finite-size numerical study of the edge mode decay. This is certainly commendable. Nonetheless, the discussion of these results is incomplete and it generates several further issues rather than clarifying the aforementioned questions.

Issue 1: The author states: "When $\alpha< 1$ the slopes exhibit
significant variations with site number $l$, however, the
variations become less pronounced with increasing chain
length (compare the blue dotted and solid curves, corresponding to L = 10000 and L = 20000, respectively)". The poor convergence of the numerical data does not come as a surprise to me, since strong long-range couplings $\alpha,\beta<1$ are known to generate large finite size corrections. Yet, the numerical curves in Fig.1 clearly do not converge to a constant slope and the author only presents the study of a single size (apart from 1 set of values where both $L=10^{4}$ and $L=2*10^{4}$). This is far from being enough to argue convergence of the numerics to the analytics. I suggest the author to present some form of finite size scaling for various (exponentially increasing) system sizes 2^10, 2^{12}, 2^{14}, and so on, to show that the central slope is converging to the expected value of Eq. (28).

Issue 2: While I can live with some finite size corrections at $\alpha,\beta<1$, the black curve in Fig.1 with $\alpha=1.5$ and $\beta=3$ also does not reproduce the result in Eq. (28). Worst of all, the curve seem to converge to a fixed slope, but the slope does not coincide with the theoretical prediction. On this matter the author only comments: "The curve corresponding to $\alpha = 1.5$ exhibts a relatively stable slope, which is somewhat smaller than the one predicted analytically [Eq. (28)]", but this is far from being sufficient. Eq. (28) was supposed to be exact (at least at $1<\alpha,\beta<2$) how can the numerics not reproduce it? Why is the result smaller than the prediction? Are those very weak finite size effects? I urge the author to present a finite size scaling for the mode decay also in this region and present convincing evidence that Eq. (28) holds in the thermodynamic limit.

Issue 3: Since the author is performing numerical analysis, I would reccommend doing so also in the region $\alpha>3$ and $\beta>2$ where the aforementioned controversy between Eq. (28) and the exponential decay took place. The author may also try some large integer power laws, e.g. \alpha=6 and \beta=4 to investigate what one may expect in the limit of the applicability of Eq. (28).

Unfortunately, in the present form the paper is not suitable to be published, but a more extensive numerical study and a softening of the current harsh claims may make it a valuable addition to the literature.

Requested changes

-) Amend/soften incorrect/harsh statements in the footnote Ref.[29].

-) Comment on the destiny of the edge mode power-law tails for large integer $\alpha$ and $\beta$.

-) Numerical finite size scaling of the edge mode decay at $\alpha,\beta<1$ to verify its convergence to Eq. (32).

-) Numerical finite size scaling of the edge mode decay at $\alpha,\beta>1$ to verify its convergence to Eq. (28).

-) Numerical finite size scaling of the edge mode decay at $\alpha=6,\beta=4$ or large values to verify the possible limitations of Eq. (28).

---

## Round 2 · Author Response

Dear Editor,

Thank you for processing my submission. I am also grateful to the Referee for their comments, especially for pointing out important references that I missed. Below I provide a list of changes in the text and the replies to the Referee’s comments.

Best regards,
Anton Bespalov

---

## Round 2 · List of Changes

Warnings issued while processing user-supplied markup:

  • Inconsistency: Markdown and reStructuredText syntaxes are mixed. Markdown will be used.
    Add "#coerce:reST" or "#coerce:plain" as the first line of your text to force reStructuredText or no markup.
    You may also contact the helpdesk if the formatting is incorrect and you are unable to edit your text.

REPLY TO THE REFEREES AND LIST OF CHANGES IN THE MANUSCRIPT

Report of the First Referee (Anonymous Report 1 on 2023-6-13)

The Referee wrote:

I have read the manuscript by the author and I was positively impressed by the in depth investigation of the edge state of the linear chain with different couplings shape. The outlined analytic method appears to be very powerful and i have no reason to doubt the correctness of the calculations. I am inclined to recommend the paper for publication provided that the author is willing to consider a few revision. The major question mark concerning the paper appears in below Eq. (33) where the author reproduces the result of Ref. [18], but the author expresses some doubts about whether this is the leading behaviour. The author shall make some major comment on what is expected to occur to Eq. (33) upon including further terms in the expansion in Eq. (A10). Indeed, the recent work [arXiv:2301.12514 ] showed how the result in Eq. (33) is not recovered by the scattering approach for α,β>2, since the leading order momentum terms of the dispersion relation change across this boundary.

Reply:

My statement concerning the asymptotic behavior of s_l in the first version of the manuscript was, probably, not very clear. In the revised version of the manuscript, Eq. (28) [formerly Eq.(33)] unambiguously indicates what the leading term looks like. For alpha>1, beta>1 and alpha≠beta the asymptotic behavior is always a power law. Including further terms in Eq. (A10) may be labor consuming, and it does not cancel the leading term. A brief discussion of this can be found after Eq. (A10). A reference to the published version of [arXiv:2301.12514] (Ref. [13]) has been added in the introduction and in the beginning of Sec. IIIA. In the published version, the authors admit that the asymptotic behavior of s_l follows a power law even for alpha>3 and beta>2, however, their approach for some reason fails to capture this. In my opinion, this approach is not mathematically justified. It relies on the assumption that the BdG equations for an infinite chain have solutions proportional to a growing/decaying exponent. However, if we substitute such coefficients u_l and v_l into Eq. (2), we obtain divergent series for the power-law model. Thus, the applicability of the scattering approach to this model is questionable. A discussion of this has been added in a footnote - currently, Ref. [29].

The Referee wrote:

The author should also acknowledge Ref. [arXiv:2211.15690] which deals on similar matters.

Reply:

This is a very important reference. This paper (currently published, Ref. [20]) has an overlap with my work in the general technique and in the part where the power-law model with alpha>1 and beta>1 is considered. I give full credit to the authors, as they obtained their results a bit earlier than me. To minimize the overlap and to enhance the novelty of my work, I made major changes to the manuscript. They are as follows: - the abstract has been rewritten; - a part of the introduction has been rewritten after the words "However, it turns out that for a wide class of systems an exact analytical solution of our Wiener-Hopf problem can be obtained..."; - Ref. [20] is now cited where appropriate; -the end of Sec. II around Eq. (20) has been extended; -the section devoted to finite-range models (formerly Sec III) has been removed, as it is covered by Ref. [20]; - results related to the power law model are now contained in Section III; - in the end of Sec. IIIC., new results concerning the numerical solution of the BdG equations in finite open chains with alpha<1 have been added. These results illustrate and complement the analytical considerations. A new figure (Figure 1) has been added. - the analysis of the model with exponential falloff of hopping amplitudes is now contained in Section IV. - in the end of Sec. V, new numerical results concerning the E_0 vs. chain length dependence in the case alpha<1 have been added. A new figure (Figure 2) has been added. - the conclusion has been reworked.

The Referee wrote:

It would be also interesting to comment on how the edge states influences other quantities aside the energy, such as the entanglement entropy of the open hand portion of the chain with respect to the infinite system. In this perspective, please notice that enhanced entanglement scaling is one of the major features of long-range Kitaev chains.

Reply:

In the revised version, the unusual scaling of the entanglement entropy is mentioned in the introduction and in the beginning of Sec. IIIC. Also, a new reference (Ref. [27]) has been added. It seems to me that the Wiener-Hopf technique and the edge states are not directly related to these interesting features. Indeed, the logarithmic entanglement scaling takes place not only in the topological phase, but also in the trivial phase (in particular, for alpha<1 or beta<1), where edge states are absent. A short remark about this has been added in the beginning of Sec. IIIC. In addition, I mention there that the power-law model with alpha<1 and beta<1 might be relevant for the description of magnetic atom chains on two-dimensional superconductors.

Report of the Second Referee (Anonymous Report 2 on 2023-7-4)

The Referee wrote:

The paper deals with a class of interesting models (Kitaev chain with long-range interactions) that, despite their apparent simplicity, displays unusual properties and a rich physics that is yet to be fully understood. The proposed rigorous analytical method is crucial to obtain important results about edge states in this model, which are carefully explained in the manuscript. The only concern is about a more precise comparison with previous results, such as the one cited in the bibliography (in particular ref. [18]) as well as more recent ones (e.g. Phys. Rev. Lett. 130, 246601 (2022), arXiv:2301.12514).

Reply:

The papers [Phys. Rev. Lett. 130, 246601 (2022)] and [arXiv:2301.12514] (currently published) are now appropriately cited, and the manuscript has been reworked accordingly - see reply to First Referee. The paper by Jäger, Dell’Anna and Morigi (currently, Ref. [19]) gives me some concerns. The analytical derivation of the Majorana wave function is contradictory: the projection operator {\cal Q} does not act according to its definition. In addition, it does not seem right that the Majorana wave function [Eqs. (34)-(36)] depends on characteristics of an arbitrary chosen Hamiltonian H_0. However, in many cases an incorrect formula for the wave function may produce correct asymptotic behavior. These issues are discussed in a footnote (currently, Ref. [29], which has been extended compared to the previous version). The numerical results from Ref. [19] look reasonable, but they are related to the case alpha>1 and beta>1, while the present paper focuses on the case alpha<1.

The Referee wrote:

This also implies to enlarge the analysis to other quantities, such as entanglement entropy and/or correlation functions that might shed light on the universal properties of the models. I understand this might be a lot of work, but at least some comments are at order.

Reply:

Calculations of correlation functions and of the entanglement entropy require the knowledge of the whole spectrum of the BdG equations. The Majorana edge mode does not seem to have a decisive influence on the entanglement entropy: indeed, the unusual logarithmic scaling occurs both in the topological and trivial phases (when edge states are absent). To calculate the mentioned quantities, a different technique should be used, as the Wiener-Hopf technique is insufficient. A remark about this has been added in the beginning of Sec. IIIC.

The Referee wrote:

The results are interesting and I would suggest publication, after requested changes are taken into account.

Reply:

I thank the Referee for their positive evaluation of my work and useful comments.

---

## Round 3 · Referee Report · Anonymous · 2023-10-3

Report

I acknowledge the fact the Author has made some changes in order to answer the Referees' comments.
However, while appreciating the analytical part of the paper, I still think that the numerical results are poor, adding very little to the theoretical calculations. Also evident discrepancies with the analytical results are not properly analysed.
Therefore, I think the paper cannot be accepted for publication as it is in SciPost physics.

---

## Round 3 · Referee Report · Anonymous · 2023-10-20

Report

I appreciated that the author clarified his viewpoint regarding the use of the scattering approach, even though the divergence referenced by the author are in the complex plane, far from the low-momentum limit $k\approx 0$ where the scattering approach is employed in Ref. [13]. Moreover, the author fails to acknowledge that (as I already mentioned) the discrepancy of results between the Weiner-Hopf (WH) treatment he employed and the scattering approach of Ref. [13] only derives from taking a different series expansion for the coefficients $\Delta_k $ and $\epsilon_k$ (and so has nothing to do with the aforementioned divergences). The author (in line with some previous literature) always keeps the non-analytic term in the expansion even when it is the largely subleading, while the authors of Ref.[13] only keep the leading contribution to the series. The author says that it is natural to keep the non-analytic contribution since they give the largest contribution in the integral (this is true both in the WH and scattering approaches), but I find it physically dubious since it yields a wildly non-universal result, which will make the study of effective models such as the Kitaev chain slightly academic. Yet, this discussion goes beyond the scope of this mathematically oriented manuscript.

More importantly, footnote [29] of the manuscript remains misleading, because the author claims the contradiction between his results and the scattering approach derives from the divergences, but it actually only derives from the use of a different expansion for the Polylogs. If one inserts the expansion used by the author in the scattering approach treatment, the power-law behaviour at all $\alpha\neq \beta$ will be obtained without the use of the WH method (and the divergence issue mentioned by the author plays no role).

The author discarded some of the previous numerical analysis because he recognized that it was misleading, but he did not make an effort to make it consistent with his analytical picture.

So, I cannot say that I am happy with all the authors replies and with all of his modifications. However, the picture delineated in the paper is now fully consistent (especially thanks to having clarified what happens for $\beta\,\& \alpha\, \in \mathbb{N}$). The manuscript contains enough relevant new information to deserve publication and the misleading statements are just marginal (thay are also extensively discussed in this thread for the interested reader). I do not intend to delay the publication process further and I give my green light.

---

## Round 3 · Author Response

Dear Editor,

Thank you for processing my submission. I am also grateful to the Referees for taking their time to study the manuscript, again. I understand that the criticism may have raised doubts about the validity of the manuscript. I hope that in the revised manuscript and in the reply below, I clarify all ambiguities and respond to all critical remarks.

The present paper provides new unique results concerning the properties of Majorana wave functions in Kitaev chains with slow power-law falloff (alpha<1) of couplings between cites. It also contributes to the resolution of the important issue whether the MZM wave function asymptotic behavior is exponential or a power law (for the model Kitaev chain under consideration): I prove analytically that it is definitely a power law in the general case (all my numerical data also confirm this). I believe that the manuscript can make a valuable contribution to the theory of topological superconductivity, and for that reason I resubmit it to SciPost Physics. Still, if you decide that the value of the manuscript is insufficient for SciPost Physics, I will settle for SciPost Physics Core.

Here is a brief overview of the changes in the manuscript: - the beginning of Sec IIIB has been slightly modified; - the discussion of Ref. [13] in footnote [29] has been modified; - Appendix A has been slightly extended around the new Eq. (A8); - the discussion of numerics in Sec IIIC has been extended significantly; new graphs have been added; - a new reference [34] has been added. All conclusions remained unchanged. A more detailed description of changes along with the reply to the Referees' comments can be found below (see List of changes).

Best regards, Anton Bespalov

---

## Round 3 · List of Changes

REPLY TO THE REFEREE AND LIST OF CHANGES IN THE MANUSCRIPT
* * *
Report of the First Referee (Anonymous Report 1 on 2023-8-3)
* * *
The Referee wrote:

The authors has carefully considered the Referees' suggestions and changed the paper accordingly. In particular, now recent literature is correctly mentioned and acknowledged. I think the theoretical and analytical calculations are now complete.
However I have some questions regarding the added part on numerical evaluations of s_l and E_0(l).
Finite size effects can be very strong (especially when alpha<1) so that it is necessary to consider very long chains and values of l far from the (right) boundary of the finite chain considered in numerics. Also the theoretical predictions holds for l large. i.e. far from the left boundary l=0. This can be clearly seen in Fig 1(a), from which one can reasonably argue that the correct range to perform any fit lies between l=40 and l=5000, at least for L=20000.
This is indeed the range considered for the evaluation of the energy E_0(l) (Fig.2), which exhibits results agreeing with the theoretical predictions. Instead, the author has not used the same procedure for s_l and I wonder why a similar fit (in the same range) is not reported for curves such as the one reported in Fig 1(a), for different values of alpha.
Also, it would be appropriate to have an estimate of the error of the fit.

Reply:

The linear fit in Fig. 1a was the first thing that I did. Visually, the points in the interval l=100..6000 seem to fit nicely on a line with slope -2.47, however, fitting smaller segments produces slopes which are significantly different from each other. In the interval l=1000..2000 the slope is -2.48, while in the interval l=4000..5000 it is -2.36 (the standard error for the slope is of the order of 1E-4 in both cases). Graphs of the local slope serve as a clear illustration of the problem. There does not seem to be a "correct" range to determine the numerical slope because of its large variations. This is most likely due to the slow convergence of s_l to the MZM wave function with increasing chain length L. I extended a bit a discussion of this in the revised manuscript (Sec. IIIC).

The Referee wrote:

Finally, I have doubts on the relevance of the behaviour of d(ln s_l^2)/d(ln l) as function of l itself -shown in Fig 1(b). This quantity is calculated locally from a quantity such as s_l that on a chain of length L is for sure affected strongly by both the finiteness of the chain itself and of the lattice spacing.

Reply:

Here, I do not fully understand the Referee. There are know issues with numerical calculations of derivatives from discrete approximations of functions, however, my mathematical model is discrete from the beginning. There are no true derivatives and no variable lattice spacing. It is not assumed that the model mimics continuous BdG equations, and it is physically relevant in the discrete form, as discussed in the introduction.
Next, for a fixed site number l the quantity s_l approaches the MZM wave function as L tends to infinity (lattice sites are numbered from 1 to L). Hence, for sufficiently large L the quantity s_l in the vicinity of the left edge of the chain exhibits very weak variations with L (and so does the local slope). This is illustrated by the finite size scaling of s_l and of the local slope, added at the request of the Second Referee — both quantities clearly converge when L is increased, though finite-size effects are indeed strong for alpha<1 (convergence is slow) — see Sec. IIIC.
* * *
Report of the Second Referee (Anonymous Report 2 on 2023-8-10)
* * *
The Referee wrote:

I read the revised manuscript from the author and I acknowledge that he made a serious effort to address mine and the other referee concerns. Unfortunately, part of these modifications are bound to generate confusion rather than clarify the open issues. In my previous report I requested the author to clarify the issue of the decay of the Majorana edge states for power-law interactions with \alpha,\beta>2 due to the mismatch between the result of the author (and of Ref. [20]) with the ones of Ref. [13]. The author, in agreement with Ref.[20], obtains the power-law decay in Eq.(28)
s_{l}\propto l^{-\delta} with \delta=min(\alpha,\beta).
Ref.[13] finds that the above result is only applicable for \alpha<3 or \beta<2, while at larger decays the approximation of Ref.[13] yields that long-range interactions are in-influential to the edge modes decay.
In the revised version of the manuscript the author claims that the result in Eq.(28) is the leading one and that all additional terms are subleading. In order to justify the discrepancy with the finding of Ref.[13] the author also makes some general claims in a footnote (Ref. [29]), where he states that Ref.[13] is "most likely erroneous". In order to sustain this latter claim the author states that assuming ul,vl∝exp(−λl) with Re(λ)>0 yields a diverging result in Eq.(2) .
Actually, it is the claim of the author to be erroneous. In fact, if one assumes the exponential decay of the BdG amplitudes with positive decay exponent the summation on the l.h.s. of Eq.(2) is convergent and, accordingly, cannot serve as an argument against the scattering approach. I advice the author to remove his fallacious statement and amend the footnote accordingly. Also, I would advise the author to be more cautious when implicating possible mistakes in other published works.

Reply:

For a semi-infinite chain, the series in the left-hand side of Eq. (2) with ul,vl~exp(−λl) with Re(λ)>0 converge, there is no doubt about it. The Referee misunderstood me. My claim is that the series diverge for infinite, not semi-infinite chains. This appears to be an obstacle for the scattering approach (see below).

To prevent further misunderstanding, let me express my concerns about Ref. [13] in more detail. The scattering approach, as described in the mentioned paper, is based on writing the MZM wave function as a linear combination of solutions of a "formal" eigenvalue problem for an infinite chain (see Appendix A in Ref. [13]). These solutions have the form of growing/decaying exponents (except for some rare cases, when they are algebraic-exponential hybrids). This procedure typically ensures that the wave function satisfies the BdG equations far from the chain edge. To satisfy these equations near the edge, appropriate expansion coefficients (and energy) should be chosen for the "formal" eigenvalue problem solutions. This method works perfectly for chains with finite-range couplings. The problem with algebraically decaying couplings is that the exponentially growing/decaying solutions do not exist in an infinite chain due to the appearance of divergent series in the BdG equations, as said above. As a result, it is not clear what the building blocks for the MZM wave function should be. In Sec. V of Ref. [13], the authors construct such building blocks by extending the expressions for Delta_k and epsilon_k in terms of Clausen functions, which were derived for real k, to complex k. However, for complex k the series in Eq. (5) diverge, so that Delta_k and epsilon_k are undefined (this is exactly the same divergence issue as mentioned above). The exponential building blocks from Sec. V do not solve the "formal" eigenvalue problem, and hence it is not clear why an arbitrary linear combination of them should satisfy the BdG equations.

To sum up, in my view, Ref. [13] contains a mathematical error, which is described above and which leads to discrepancies with other published papers [19,20]. Nevertheless, in accordance with the Referee's recommendation I soften the criticism in the footnote and make my point concerning divergent series clearer.

The Referee wrote:

More in general, it should be noted that all approaches of Ref. [13], of Ref.[20] and of the current manuscript, albeit using different methods, produce similar equations. The main discrepancy between the approaches of Ref. [13] and Ref.[20] depends on the way the polylogarithm singularities in the complex plane are treated. Indeed, Ref.[20] always keeps the non-analytic term in the expansion of the PolyLog. On the contrary, Ref.[13] only focuses on the leading order contribution, which becomes analytic for α>3 and β>2.

Reply:

I want ot make a comment here. Approaches based on the Wiener-Hopf technique (the present work and Ref. [20]) take into account both the non-analytic and analytic contributions to the PolyLog in the vicinity of the point z=1. However, it turns out that in the leading order the asymptotic behavior of the MZM wave function is always determined by the non-analytic contribution. The formal explanation of this is relatively simple. The leading non-exponential contribution to the long-range behavior of the wave function is generally determined by Eq. (A5). The function Q_+^{-1} here can have a leading analytic part, however, it vanishes after integration over the branch cut. Hence, the non-analytic part always matters, even if it is small.

The Referee wrote:

The author's result is specular to the one of Ref.[20], but the author states that Eq. (28) can only be applied to the case of non-integer α,β. This is in agreement with the aforementioned picture in which the non-analyticity of the polylog function plays a crucial role in the Wiener-Hopf method. However, this result leaves open two alternative scenarios:
a) For integer power-laws e.g. α=6 and β=4, the decay of the modes is still power-law.
b) The power-law behaviour only occurs at non-integer values and the integer case displays exponential behaviour.
The author stated that his analysis unambiguously identifies Eq. (28) as the leading behaviour for the decay of the edge mode, but what scenario is to be expected? In the case (a) the author shall clarify how the power-law behaviour is recovered in absence of the polylog branch cut. In the case (b) the result will be quite unphysical since all integer power-law decays will behave differently from the non-analytic case. These question is not physically irrelevant since most of the naturally occurring power-law potentials display integer power-laws (gravity, coulomb interaction, dipolar, van-der-Wals,...).

Reply:

I extended the calculations to integer alpha and beta, which made the analysis more complete. When alpha and beta are integers, the polylog still has a branch cut (in this respect, Eq. (32) from Ref. [13] does not work for integer alpha and beta). This should not be surprising, because the polylog of order 1 is a logarithm. The behavior of polylog close to z=1 is given by the newly added Eq. (A8) (unfortunately, I was unable to find this relation in a textbook, so the technical report [34] is cited, which is available online: https://www.cs.kent.ac.uk/pubs/1992/110/content.pdf). All other equations in the manuscript remain the same for integer alpha and beta greater than 1. Accordingly, the beginning of Sec. IIIB has been modified.

The Referee wrote:

In the revised version of the manuscript the author also includes some finite-size numerical study of the edge mode decay. This is certainly commendable. Nonetheless, the discussion of these results is incomplete and it generates several further issues rather than clarifying the aforementioned questions.
Issue 1: The author states: "When α<1 the slopes exhibit significant variations with site number l, however, the variations become less pronounced with increasing chain length (compare the blue dotted and solid curves, corresponding to L = 10000 and L = 20000, respectively)". The poor convergence of the numerical data does not come as a surprise to me, since strong long-range couplings α,β<1 are known to generate large finite size corrections. Yet, the numerical curves in Fig.1 clearly do not converge to a constant slope and the author only presents the study of a single size (apart from 1 set of values where both L=104 and L=2∗104). This is far from being enough to argue convergence of the numerics to the analytics. I suggest the author to present some form of finite size scaling for various (exponentially increasing) system sizes 2^10, 2^{12}, 2^{14}, and so on, to show that the central slope is converging to the expected value of Eq. (28).

Reply:

I added the finite-size scaling for s_l and for the local slope — see the new Fig. 1 and extended discussion in Sec. IIIC. Both quantities seem to converge with increasing L, but convergence is very slow. Finite-size effects are strong, indeed. Arguably, the local slope [Eq. (34)] for L=20000 matches the local slope of an isolated Majorana mode in the range l<200, which seems insufficient to reach the limit of large l, where Eq. (32) works well. Of course, this feels unsatisfactory if we want quantitative comparison with Eq. (32), however, the key features of this equation are reflected by the numerical data. In particular, the local slope lies between -2 and -3 (for not too small site numbers), and it increases with increasing alpha, unlike in the case alpha>1.
On the other hand, the peculiar results for finite chains are interesting from a practical perspective: experimental systems described by the discrete BdG equations (e.g., magnetic atom chains on superconductors) are finite and, in fact, not very long.

The Referee wrote:

Issue 2: While I can live with some finite size corrections at α,β<1, the black curve in Fig.1 with α=1.5 and β=3 also does not reproduce the result in Eq. (28). Worst of all, the curve seem to converge to a fixed slope, but the slope does not coincide with the theoretical prediction. On this matter the author only comments: "The curve corresponding to α=1.5 exhibts a relatively stable slope, which is somewhat smaller than the one predicted analytically [Eq. (28)]", but this is far from being sufficient. Eq. (28) was supposed to be exact (at least at 1<α,β<2) how can the numerics not reproduce it? Why is the result smaller than the prediction? Are those very weak finite size effects? I urge the author to present a finite size scaling for the mode decay also in this region and present convincing evidence that Eq. (28) holds in the thermodynamic limit.

Reply:

For the black curve, the difference between the theoretical and numerical slopes is smaller than 0.4% for site number l=5000, and it becomes even smaller closer to the center of the chain. I would not argue that there is a contradiction with theory here. The asymptotics for s_l contain sub-leading terms, which become more significant as alpha and beta approach 1, and which cause deviations from simple power-law behavior (still, for sufficiently large l the power law is always recovered). Unfortunately, these sub-leading terms are very hard to evaluate.
To clear up the ambiguity mentioned by the Referee (which is connected with somewhat poor presentation of the numerics rather than a fundamental issue), I remove the discussion of the curve with alpha=1.5 (and the curve itself — Fig. 2 has been modified), and instead concentrate on a more extended discussion of the case alpha=6 and beta=4 (with finite-size scaling) — see below.

The Referee wrote:

Issue 3: Since the author is performing numerical analysis, I would reccommend doing so also in the region α>3 and β>2 where the aforementioned controversy between Eq. (28) and the exponential decay took place. The author may also try some large integer power laws, e.g. \alpha=6 and \beta=4 to investigate what one may expect in the limit of the applicability of Eq. (28).

Reply:

I added the numerical analysis for alpha=6 and beta=4 — see Fig. 3 and the end of Sec. IIIC. Without doubts, the asymptotic behavior of the edge state follows a power law, and it is in good agreement with Eq. (28), which holds also for integer alpha and beta, as said above.

The Referee wrote:

Unfortunately, in the present form the paper is not suitable to be published, but a more extensive numerical study and a softening of the current harsh claims may make it a valuable addition to the literature.

Reply:

I thank the Referee for their recommendations and incorporate the requested corrections.

---

## Editorial Decision

published